# Oxygen Permeability of Silk Fibroin Hydrogels and Their Use as Materials for Contact Lenses: A Purposeful Analysis

**DOI:** 10.3390/gels7020058

**Published:** 2021-05-11

**Authors:** Traian V. Chirila

**Affiliations:** 1Queensland Eye Institute, South Brisbane, QLD 4101, Australia; traian.chirila@qei.org.au; Tel.: +61-(0)7-3239-5024; 2School of Chemistry & Physics, Queensland University of Technology, Brisbane, QLD 4001, Australia; 3Australian Institute of Bioengineering & Nanotechnology (AIBN), The University of Queensland, St Lucia, QLD 4072, Australia; 4Faculty of Medicine, The University of Queensland, Herston, QLD 4006, Australia; 5School of Molecular Science, The University of Western Australia, Crawley, WA 6009, Australia; 6Faculty of Medicine, George E. Palade University of Medicine, Pharmacy, Science & Technology, Târgu Mureş 540139, Romania

**Keywords:** silk fibroin, membranes, biomaterials, oxygen permeability, contact lens

## Abstract

Fibroin is a fibrous protein that can be conveniently isolated from the silk cocoons produced by the larvae of *Bombyx mori* silk moth. In its form as a hydrogel, *Bombyx mori* silk fibroin (BMSF) has been employed in a variety of biomedical applications. When used as substrates for biomaterial-cells constructs in tissue engineering, the oxygen transport characteristics of the BMSF membranes have proved so far to be adequate. However, over the past three decades the BMSF hydrogels have been proposed episodically as materials for the manufacture of contact lenses, an application that depends on substantially elevated oxygen permeability. This review will show that the literature published on the oxygen permeability of BMSF is both limited and controversial. Additionally, there is no evidence that contact lenses made from BMSF have ever reached commercialization. The existing literature is discussed critically, leading to the conclusion that BMSF hydrogels are unsuitable as materials for contact lenses, while also attempting to explain the scarcity of data regarding the oxygen permeability of BMSF. To the author’s knowledge, this review covers all publications related to the topic.

## 1. Introduction

The major component of the silk thread produced by the larvae of domesticated silk moth (*Bombyx mori*), and of wild species, is a protein known as fibroin. This biopolymer is a naturally designed polypeptidic composite belonging to the group of fibrous proteins. The outstanding functional performance of the silk thread is due to the composition of fibroin, which consists of highly repetitive amino acid sequences able to induce a dominant homogeneous secondary structure, based on β-strands mutually linked through amide–carbonyl hydrogen bonds that lead to the most robust interstrand stability. 

The extensive literature regarding the applications of silk fibroin, especially of *B. mori* silk fibroin (henceforth, BMSF), in biomedicine and bioengineering has been illustrated in a number of seminal reviews [1,2,3,4,5,6,7,8,9,10,11,12,13,14,15]. The BMSF hydrogels display features that make them attractive as biomaterial substrates (films, membranes, coatings, fibrous, or porous scaffolds) for cell attachment and growth with an aim to regenerate tissues. These features include acceptable mechanical strength and tissue-like compliance, minor inflammatory responses, suitable permeability for the nutrient/waste cellular exchange, protracted biodegradability, and―when required―transparency. BMSF has been proposed and evaluated as a membranous substrate for the cells of the eye (corneal, retinal) in ophthalmic tissue engineering and regenerative ophthalmology studies [4,13,16,17,18,19,20,21,22,23,24,25,26,27], albeit some concerns have been recently discussed [28]. In such applications, suitable transport properties are necessary in order to ensure the presence of oxygen and nutrients at a level that is relevant to the growth, proliferation, and viability of cells, and to the production of extracellular matrix. Of main interest to my analysis is the fact that BMSF hydrogels have been also proposed as materials for manufacturing contact lenses, where it is well known that an enhanced permeability to oxygen is paramount. While the level of the oxygen permeability may be sufficiently high for certain tissue engineering applications of BMSF, clearly it would not be enough for contact lenses. In ignorance of this recognized and important fact, suggestions to make contact lenses from BMSF hydrogels still emerge sporadically in the literature. A few years ago, I have reviewed [29] the literature related to the evaluation of oxygen permeability of BMSF and discussed critically the issue of using BMSF as a material for contact lenses. That review was published in a journal that is currently defunct. The present review is an extended, updated, and substantially rewritten version of that article.

Compared to other properties of BMSF, the oxygen permeability has been much less investigated, and the results are generally controversial. There have been relatively few researchers involved in measuring, or in attempting to improve, the oxygen permeability of BMSF hydrogel membranes. This review critically analyzes those efforts, while also showing that, by today’s standards, the oxygen permeability of BMSF hydrogels is not adequate for manufacturing contact lenses. 

## 2. The Oxygen Permeability of Silk Fibroin and The Issue of Contact Lenses

Many of the publications on the biomedical applications of BMSF point out that this material is permeable to oxygen [29]. Besides, statements that BMSF hydrogels can be used as materials for making contact lenses are also present in some articles. Generally, no sources are cited to support such assertions; when, however, references are given, most of the reports cite the publications of Minoura’s group in Japan, widely recognized as being the first investigators to introduce and evaluate BMSF as a biomaterial [30,31,32]. They have used in-house previously developed electrochemical methods and custom-made instrumentation [33] to measure the oxygen permeability (henceforth, ***P***) of BMSF hydrogel films, and have concluded that the measured values situate BMSF among the materials suitable for contact lenses [30]. In reality, the highest value that they measured at physiological temperatures was around 10 Barrer for those BMSF films that were treated in aqueous methanol (a process leading to physical crosslinking and ensuing gelation, in fact a structural stabilization by conversion to the silk II conformational polymorph). The equilibrium water content of the membranes was between 20% and 40% and decreased with increasing duration of immersion in methanol. The measurement of ***P*** was done “in wet membranes”, but there is no indication as to how the dehydration of the hydrogel film was avoided during measurements. As a note, the “Barrer” is the accepted name for the non-SI oxygen permeability unit, used extensively in contact lens industry, and defined [34,35] as:10^−10^(cm^3^O_2_(*STP*)·cm)/(cm^2^·s·cmHg), or 10^−11^(cm^3^O_2_(*STP*)·cm)/(cm^2^·s·mmHg),
where “STP” stands for “standard temperature and pressure” (and is not always mentioned). Expressed in SI pressure units, 1 Barrer = 1330 (cm^3^O_2_·cm)/(cm^2^·s·Pa).

In a further study [36], Minoura and colleagues reported values for ***P*** of BMSF between about 3 and 11 Barrer and investigated in more detail the effect of the immersion time in methanol. The shorter the immersion was, the higher both water content and permeability were. A “permeability coefficient” ***P***_a_ has been also introduced to represent the oxygen permeability of the totally amorphous fibroin. For the BMSF hydrogel film with the highest water content, a value of ***P***_a_ ~3 Barrer has been estimated. Although the oxygen permeability values measured by Minoura and colleagues might have been comparable to the values of ***P*** (or ***Dk*** in contact lens terminology) of some daily wear contact lenses available at that time on the market, a value of 10 Barrer is definitely unacceptably low by the current standards. Such a low oxygen permeability cannot allow sufficient oxygen to reach the ocular surface and assure the normal metabolism of the cornea during contact lens wear [37,38,39]. For the contemporary contact lenses, values of ***P*** for daily wear are commonly over 60 Barrer, while for extended wear is over 100 Barrer and can be as high as 140 Barrer and beyond. To put into a larger perspective between two extreme limits, ***P*** for polydimethylsiloxane is 600 Barrer [40], and for poly(methyl methacrylate) (henceforth, PMMA) is 0.5 Barrer [39]. 

In the tissue engineering applications, too little oxygen passing across a BMSF membrane into the physiological environment of cells can be detrimental to cellular viability and, consequently, to the development of suitable substrates in the fibroin-cell constructs. We cannot share some investigators’ enthusiasm unleashed by a ***P*** around 10 Barrer for the BMSF membranes. If nothing else, such a value denotes poor oxygen permeability, a situation that can only lead to insufficient oxygen available for the cellular metabolism. However, I have to emphasize that, in the case of BMSF, this aspect is much less critical for cell substrates than it is for contact lens materials.

Despite the available information on oxygen permeability requirements that Minoura and colleagues would have had access to, they were confident enough to apply for, and obtain, patents related to the use of BMSF [41] or its blends with some synthetic polymers [42,43] as materials for contact lenses. Measured values of ***P*** disclosed in these patents include 10.6 to 13.3 Barrer for BMSF as such [41], and 10 to 25 Barrer for blends of BMSF with certain vinyl polymers [43]. 

While there is no evidence that Minoura’s proposal has ever led to a marketable product, the idea that contact lenses can be made from BMSF appears to stay with us. For instance, such an application is mentioned in a review [44], citing Minoura’s patents but without discussing whether these patents have ever been implemented. In a more recent publication [45], the authors backed up the idea of BMSF contact lenses by citing an article [46], which in reality was a study on enzymatic control of β-sheet secondary structure in genetically engineered silk-like proteins, therefore it does not have any relation to the topic of contact lens, and clearly invalidates the claim in the context. About a decade ago, a rather peculiar development involved Tufts University in Boston, USA, which houses the world’s premier centre for silk research. In several internet media releases, it was disclosed that their scientists are developing “new silk-based contact lenses” as “a nontoxic alternative to glass and plastics”, and that they are edible too [47,48], without any mention of the crucial need for oxygen permeability. Apart from the fact that glass or polymers intended for biomedical application are on purpose selected not to be toxic, and the fact that glass is no longer used in contact lens manufacture, the idea of eating your own contact lens does not offer any rational advantage and is pointless, if not plainly risible. Auspiciously (and conspicuously too), most of the media releases regarding this development, including a Discovery Channel presentation, have been removed in the meantime from the web.

## 3. Further Investigations on Silk Fibroin Oxygen Permeability 

It was not until a decade after Minoura’s reports that other investigators undertook further assessment of the oxygen permeability of BMSF hydrogel membranes. Researchers at Seoul National University have evaluated blends of BMSF with chitosan [49], with an aim to use such membranes as artificial skin and wound dressings. The oxygen permeability was measured in a custom-made two-compartment diffusion cell equipped with an oxygen sensor. The value of ***P*** for BMSF alone was around 0.25 Barrer, while for the blend 50% BMSF + 50% chitosan it was around 0.58 Barrer. These surprisingly low values indicated that practically the materials allow minimum oxygen transport, which has not deterred the authors from stating [49] that the blends “showed very high oxygen permeability”, despite their measured ***P*** of the BMSF hydrogel being even less than that of PMMA, a glassy polymer that is virtually considered as impervious to oxygen. It is not clear how the evaporation of water from the hydrogel membranes (estimated to have an initial water content of 33% on hydrated basis [49]) has been prevented during measurements. Besides, it is hard to believe that by mixing two different materials that each have poor oxygen permeability, the result is a mixture that acquires a permeability higher than any of its components. As I have suggested before [29], the factuality of some of the Korean group’s conclusions may be questionable, however the very low ***P*** values are plausible.

A more recent study [50] has been carried out at Tufts University. They have investigated the effects of water annealing and of treatment with aqueous methanol on some characteristics of BMSF hydrogel membranes including the oxygen permeability. For measurements, a commercially available oxygen permeation analyzer was used to provide the values for the oxygen transmission rate (OTR), which were then converted into ***P*** values (in Barrers). During measurements, the relative ambient humidity was maintained at two levels of 50% and 80%, respectively. It was found that the oxygen permeability of BMSF membranes treated with methanol was higher that than that of the water-annealed membranes, and that both relative humidity and duration of treatments had a marked effect on the values of measured ***P***. It appears that the water content of the water-annealed BMSF films was nil (linear swelling ratio *Q* = 1), while the methanol-treated films retained water (*Q* = 1.6). (We should note that in the absence of numerical data for the densities of dry and hydrated BMSF, the value of *Q* cannot be converted into percentage water content [29].) For the water-annealed films, ***P*** was between 0 and 1.8 Barrer, while for the methanol-treated ones was between 0.25 and ~5 Barrer. The differences have been attributed to changes in the secondary structure of fibroin caused by the two different treatments [50]. Water annealing induced a more densely packed β-sheet conformation as compared to the less ordered packing induced by the treatment with methanol. These properly conducted experiments proved convincingly that the oxygen permeability of BMSF is low. The same investigators reported later [51] that a low permeability of BMSF is advantageous when it is used as a coating for perishable food, e.g., fruits and vegetables, which require an optimal preservation of freshness during storage. Depletion of oxygen reduces metabolic activity and causes less decay of the fruits or vegetables, thus enhancing their shelf life.

Another confirmation of the inherently low oxygen permeability of BMSF has been provided in a master’s degree thesis [52] presented at the University of Waterloo in Canada. A custom-made permeation cell setup has been used for measuring the permeation of common gases (O_2_, N_2_, CO_2_) through the BMSF hydrogel membranes. The water content in the membranes was expressed as a degree of swelling of 473%, which is equivalent to around 80% when expressed as water content on a hydrated basis. The drying of membranes during measurements was prevented by purging the permeation cell with a stream of humidified oxygen, but we do not know how effective this method was. The oxygen permeability of BMSF hydrogel film was found to be 5 Barrer [52].

## 4. Oxygen Permeability of Silk Fibroin in Regenerative Ophthalmology 

As a membranous substrate for cell growth, a BMSF hydrogel is expected to possess transport and mechanical properties superior to the human amniotic membrane (henceforth, AM) if is intended to replace the latter. The amniotic membrane is essentially a basement membrane constituting the innermost layer of the placenta and is harvested from donor mothers at birth during selective caesarean sections. Transplantation of AM is regarded as a salient element of the major surgical procedures for the management of ocular surface diseases [18,53,54]. A comparison between the oxygen permeability of AM and that of BMSF would therefore be pertinent.

My previous review [29] and subsequent literature searches indicated that there is only one published estimation of the oxygen permeability of AM [55]. Regrettably, the authors chose to calculate the value of ***P*** using an equation, instead of measuring it experimentally. Equations of the form ***P*** = A*e*^B*W*^ are indeed available for calculation of the oxygen permeability of hydrogels, such as Fatt equation [56] where A = 2 and B = 0.0411, or Morgan-Efron equation [37,57] where A = 1.67 and B = 0.0379. In these equations, *W* is the equilibrium water content at room temperature of the hydrogel material, *e* is the base of the natural logarithms, while A and B are constants determined experimentally from the measured *W* and ***P*** of common synthetic hydrogels used for manufacturing contact lenses. Apart from inherent drawbacks related to the use of such equations, as discussed by Tighe and Mann [58], here we have to encounter an additional problem, i.e., that using this equation for calculating the oxygen permeability of AM may not be justified. First, the equations can be applied only to synthetic polymeric hydrogels (mostly carbon-backbone polymers), as these equations are based specifically on the experimentally measured characteristics of such hydrogels. The structure and composition of synthetic hydrogels is vastly different from those of biological tissues or individual proteins such as silk fibroin. There should be little or no expectation that a biological hydrogel within our body, which was designed by nature to fulfil a complex evolutionary task, would display the same transport properties as an artificial material only because they both may have coincidentally the same water content. Second, the authors used a wrong equation where A = 2.667 [55], while further stating that calculation was done according to the international standard ISO 9913-1. If so, this was incorrect: that particular standard [59] clearly recommends the use of Fatt equation where A = 2; a value of 2.667 for the coefficient A cannot be found in any of other published equations and potentially is erroneous. In addition, the same standard [59] recommends that the equation should be applied only to the hydrogel designated as material for normalization, in this case poly (2-hydroxyethyl methacrylate). Moreover, the standard ISO 9913-1 has been long withdrawn and superseded by ISO-18369-4 [60], which does not recommend the use of equations.

It can be argued that AM might have a ***P*** close to that of collagen, since this membrane is rich in this protein. However, there are some issues of concern. First, the structure and composition of AM are both much more complicated than those of collagen, comprising a cellular epithelium, a collagenous basement membrane and a stroma that is itself composed of other three layers (the compact, fibroblast and spongy layers) [54]. The calculation of ***P*** for multilayer membranes requires a different approach, which involves complex equations [61], an aspect ignored by these investigators [55]. Second, the literature on the collagen oxygen permeability is itself replete with conflicting results. While in some commercial collagen bandage lenses (which are designed to last only a few days onto the ocular surface and then dissipate) values of ***P*** = 11.5–21.8 Barrer [62] and ***P*** = 26 Barrer [63] have been respectively reported, other investigators found that collagen films had such a low oxygen permeability [64] that they would certainly be suitable for food packaging [65,66].

In conclusion, the value ***P***~143 Barrer for the amniotic membrane reported by these authors [55] has resulted from a calculation based on an incomplete and partially wrong algorithm and an obsolete ISO standard, has not been obtained by experimental measurements, it is based on flawed evidence, and therefore cannot be considered as a true value for the oxygen permeability of AM. In addition, in the hydrogels where water is the dominant vehicle for the transport of oxygen molecules, there is a limiting value for ***P*** of 100 Barrer, which is the theoretical value for pure water (i.e., for a hypothetical material with the water content of 100%) [58]. This factor alone would invalidate the value reported in the study if we accept that water contributes essentially to the oxygen transport mechanism, which is a highly probable scenario [29].

Following a similar methodology, other ophthalmic investigators have reported [67] the calculation of oxygen permeability for “four varieties of silk films”, presumably to be used as substitutes for AM in regeneration of ocular surface. The nature of the samples was not disclosed; however, based on other reports from the same authors, and considering that some of the samples were sourced from my laboratory, we can be sure that the materials were BMSF hydrogel films. The authors specified that their calculation was based on the water content (*W*) of the films, and that it has been done according to an international standard, which clearly indicates that they followed the procedure mentioned above [55], however without mentioning what equation has been used for the calculation of ***P***. For two of the sample groups, a value of ***P***~100 Barrer was calculated, while for the other two, ***P*** = 14 Barrer and ***P*** = 27 Barrer were respectively obtained [67]. As expected for a short conference abstract, no other details of samples and measurements were provided. These results appear as unreliable as those reported for the amniotic membrane [55], and therefore should be disregarded.

Our group at the Queensland Eye Institute, in Brisbane, Australia, was the first to propose and assess the BMSF hydrogels as biomaterial substrates for the culture of various cells of the eye [4,16,17,18,19,20,21], and a variety of BMSF membranous substrates for corneal and retinal cells have been prepared and evaluated in our laboratories aiming at developing cell therapies for eye diseases. On the background of the controversial values reported in the existing literature for the oxygen permeability of BMSF, we decided to investigate this aspect. For the present study, two 30-μm thick different membranes were prepared. The membrane “A” was obtained by a standard protocol [17,32], then subjected to water annealing, and finally treated with ethanol to become a gel. The protocol for membrane “B” included two additional steps, a treatment with poly(ethylene glycol) (henceforth, PEG) to generate a porous morphology [68], followed by enzymatic crosslinking (with horseradish peroxidase) for increasing the mechanical strength [69]. The oxygen permeability of these membranes has been measured at a specialized service laboratory in Japan (Kureha Special Laboratory Co. Ltd., Iwaki, Japan), in an OX-TRAN Model 2/21 System (Mocon, Inc., Minneapolis, MN, USA), the method being based on coulometric diffusion [60]. For the membrane “A”, a value of ***P*** = 0.61 Barrer was measured, while the more porous membrane “B” had ***P*** = 1.98 Barrer [69]. These values are considerably lower than the previously values obtained by calculation [67].

## 5. Oxygen Permeability of Fibroin-Based Blends

The blending of BMSF with other polymeric materials is an alternative strategy to improve certain physical properties. Chitosan appears as a preferred component in the blends proposed as materials for contact lenses. In addition to the report discussed above [49], other investigators have undertaken [70] the measurement of oxygen permeability of a contact lens cast from a blend of 70% chitosan and 30% BMSF, without comparing to pure BMSF. The lens was made by the cast spinning method and had a central thickness of 0.2 mm. The measurements were performed according to the current ISO standard [60], by the polarographic method. They found a value of 26 Barrer, which is 45 times higher than the value reported [49] by a Korean group for a similar blend (50% chitosan and 50% BMSF). Such a discrepancy may be due to differences in the experimental design. For chitosan as such, a value of 22 Barrer was measured in the same conditions [70]. These results raise again the question as to how a mixture can acquire an oxygen permeability higher than any of its components, as obviously each component had lower oxygen permeability. In a subsequent study [71], the investigators proposed their chitosan-BMSF blend as a material for therapeutic contact lenses able to deliver ocular dugs. This suggestion is fraught with problems. First, the authors have suggested daily-wear contact lenses, while it is known that the therapeutic contact lenses must assure a prolonged delivery of drugs to the ocular surface, therefore extended-wear contact lenses would be preferred. Second, such lenses must have high oxygen permeability [72,73], either for daily or for extended wear. As an example, three therapeutic contact lens brands currently successful on the market, Acuvue Oasys™ (Vistakon), Pure Vision™ (Bausch and Lomb), and Air Optix Night and Day Aqua™ (Alcon) have a ***Dk*** (***P***) of 103 Barrer, 99 Barrer, and 140 Barrer, respectively [72]. Even if we accept the rather dubious value of ***P*** = 26 Barrer for the chitosan-BMSF blend as reported previously [70], such a permeability to oxygen would still not be suitable for the proposed application: by the time the drug would be completely released, the cornea would become dysfunctional due to oxygen deprivation.

My further literature search showed that a group at Chiang Mai University in Thailand have carried out oxygen permeability measurements for blends containing BMSF as a minor component [74,75]. A rather rudimentary procedure, the differential pressure method (the “manometric” method), has been used to measure indirectly ***P***, which consequently was expressed in percent ratio between two differential readings on the U-tube of the manometer scale. This is a non-dimensional unit (% cm/cm) that is not convertible into Barrer or other conventional units [29]. Blends containing 2% wt/vol BMSF, or less, with poly(vinyl alcohol) and rice starch have been studied [74]. The OTR values have been also measured in a commercially available device, but the results were not converted into units for ***P***. Both ***P*** (in % cm/cm) and OTR values indicated that oxygen permeability was the highest at a content of 2% wt/vol BMSF. However, if OTR values measured in this study [74] are compared with those reported elsewhere [50] for BMSF, the inescapable conclusion is that the oxygen transport through the blended membranes was extremely low. The same group also investigated [75] blends consisting of 5% wt/vol BMSF, rice starch and trisodium trimetaphosphate (The last component was added as a crosslinking agent for the starch). Additionally, the materials were rendered porous by a freeze-drying process. As OTR has not been measured in this study, a discussion on ***P*** values based on simple readings on a manometric scale is meaningless, and the observed trend of ***P*** to increase with increasing porosity is something to be expected. However, whether or not the levels attained would be physiologically suitable in a biomedical application cannot be asserted from these reports.

## 6. Effect of Porosity

Further investigations on the role of porosity on the oxygen permeability of BMSF have been carried out by the same group at Chiang Mai University, this time using hydrogels containing BMSF only [76]. Porosity has been induced by adding PEG as a porogen, a well-known method. In this study, PEG with a molecular mass of 400 kDa has been used. Membranes were also made by the chemical crosslinking of BMSF with glutaraldehyde, a chemical compound that is not known as a porogen. The authors justified the use of chemical crosslinking as a tool to induce pores in BMSF by an assumed ability of glutaraldehyde “to create more empty spaces within the membrane” [76], a statement suggesting that the authors may possess an incomplete understanding of the mechanism of crosslinking processes in polymers. ***P*** was expressed in percent ratio read on a manometric scale. When it came to a relation between porosity and ***P***, these authors have found that both properties presented maxima at 40% wt PEG and at 3% wt glutaraldehyde, respectively. The drops in permeability recorded prior and after the maxima have been clumsily explained [76] by “more crosslinking between PEG and SF chain when they come together” and, respectively, by the fact that “the membranes become more dense due to extensive cross-linking”. Neither explanation is intelligible. The results of this study, as well as of the other reports coming from the same group [74,75], are rather detrimental to a better understanding of the oxygen transport through the BMSF hydrogel membranes, and scientifically meaningless.

## 7. Summary and Conclusions

Table 1 presents a summary of data reported on the oxygen permeability of BMSF. The table contains only those data that can be considered reliable and mutually comparable.

In spite of some variability, the data in Table 1 indicate unequivocally that BMSF hydrogels display inherently a low permeability to oxygen.

The paucity of reported data is a reflection of the difficulties related to the estimation of oxygen permeability of BMSF hydrogels, rather than being caused by lack of importance of, or lack of interest for, this aspect of silk research. Based on my previous analysis [29], the factors that can affect negatively the availability, accuracy and reproducibility of information related to the topic of BMSF oxygen permeability are listed below. I shall mention that some of these factors are not specific to the BMSF hydrogels.

(a) Variability of the procedures and instrumentation from one laboratory to another: virtually, identical methods are seldom employed in different published reports, although the most recent standard, ISO 18369-4 [60], specifically recommends the polarographic method for all types of materials (including hydrogels), and the coulometric method for non-hydrogel materials.

(b) The instruments available commercially for measuring oxygen permeability are too costly for most of the (usually) small silk research laboratories.

(c) There may be variability in the protocols to isolate the silk fibroin from silk cocoons and obtain hydrogels. Slight differences between protocols can affect significantly the structure and properties of the regenerated fibroin.

(d) The use of many units different from the Barrer, sometimes impractical hybrids of SI, imperial and US units, which makes the comparison between the results of the measurements performed in different laboratories very difficult, if not impossible.

(e) The problem above is due in part to the use in some laboratories of unsuitable methods to measure ***P***. Generally, such methods are associated with the need to introduce improvised units that are neither convertible into Barrer, nor meaningfully interpretable. Moreover, when contact lens hydrogels with a known ***P*** (always provided in Barrer units by the manufacturers) are subjected to measurements in makeshift experimental setups, the results do not always match the values measured by using proper devices and standardized methods.

However, regardless of the protocols for the preparation of samples and for their measurement, the undisputable conclusion from my analysis is that silk fibroin membranes display poor oxygen transport characteristics. There is a distinct possibility that some samples of BMSF hydrogels in the studies analyzed here might have been substantially dehydrated during the permeability measurements, leading to abnormally low values for ***P***. In other words, if we accept that the water content governs, at least in part, the oxygen transport in the natural biopolymers (as it does in the carbon-backbone synthetic polymeric hydrogels), then we may posit that the values of ***P*** reported for BMSF could have been unforeseeably affected by unintended dehydration of samples. However, even so, based on those extant reports that can be regarded as reliable (see Table 1), it is evident that BMSF hydrogels do not have a sufficiently high oxygen permeability to be qualified as suitable for manufacturing contact lenses. Statements opposite to this effect should be regarded as misinformation and—for the sake of scientific rigor—the claims that BMSF hydrogels are suitable for contact lenses should not be taken at face value. We are dealing here with what is known as an *ad populum* fallacy, i.e., based on the idea that if we accept that many hydrogels are suitable as materials for contact lenses, then we shall accept that all hydrogels (including BMSF hydrogels) are also suitable, ignoring the fundamental aspect of a high oxygen permeability. The purpose of my analysis was to refute such a fallacy, and hopefully to prevent its further dissemination.

## Figures and Tables

**Table 1 gels-07-00058-t001:** Reported values for the oxygen permeability of *Bombyx mori* silk fibroin membranes.

Reported Values *P* (Barrer)	Method/Instrument	References (Year)
3–11	Electrochemical ^1^	[30,36] (1990)
10.6–13.3	n. a. ^2^	[41] (1993)
0.25	Diffusion cell/oxygen sensor	[49] (2001)
0–1.8 ^3^	OTR analyzer	[50] (2010)
0.25–5 ^4^	As above	
5	Permeation cell/flowmeter	[52] (2013)
0.61 ^5^	OTR analyser/coulometry	[69] (unpublished)
1.98 ^6^	As above	

^1^ In-house setup. ^2^ Not disclosed in the patent specification. ^3^ Water-annealed membranes. ^4^ Methanol-treated membranes. ^5^ Standard BMSF membrane, ethanol-induced gelation. ^6^ Treated with PEG and enzymatically crosslinked.

## Data Availability

Not applicable.

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
