# Peer review of "Oxygen Permeability of Silk Fibroin Hydrogels and Their Use as Materials for Contact Lenses: A Purposeful Analysis"

_gels, 2021, doi:10.3390/gels7020058_

Round 1
Reviewer 1 Report
Overview:
The author provides an extensive review of studies to characterize the oxygen permeability in silk fibroin hydrogels to determine their suitability for use as contact lenses. The manuscript is generally detailed and well-written. The primary concerns about the paper are that: (1) there are a number of instances detailed below where additional background and detail would better place the referenced work in better context; and more importantly, (2) the tone of the paper is oftentimes condescending and disparaging of other authors’ work detracting from the scientific objectivity of the analysis, especially the use of demeaning language such as “clumsily,” “bizarre,” “regrettably,” etc, and casual use of grammatical symbols, ie (!), (?), etc. Further, the first-person tone with the use of “I,” “me,” etc suggest the tone of an editorial or perspectives piece. As as an analytical review paper, such comments and tone should be rephrased to be more objective. The following comments are offered to strengthen the quality of the final manuscript and provide a review that is useful to the field rather than needlessly insulting it.
General Comments:
End of Page 3: I am not sure that is necessary to highlight that the author’s previous review was published in a now defunct journal.
Page 4: It would be helpful to describe/translate the Barrer unit into standard SI units.
Page 5, first sentence: “shortest” should be “shorter”
Page 5, second paragraph: Recommend moving the information about commercial contact lens permeability earlier in the text. Provide context for why polydimethylsiloxane and PMMA values are described. Also, it does not seem that a cited physiological requirement for a threshold/minimum permeability is provided to more objectively assess candidate materials, but rather only a comparison to commercial products.
Page 6: The following line is inappropriate and lacks objectivity: “but probably
the investigators have become confused regarding the definition and handling of the units
for permeability.” It is possible that the authors may have had a very rational reason for selecting there units, perhaps as a consequence of their experimental set-up.
Page 7: For the phrase, “I have suggested before [29], the factuality of some of the Korean group’s conclusions may be questionable, however the very low P values appear verisimilar” again the tone is condescending and the characterization is not fully explained and supported. Also, the word “verisimilar” does not seem to be the right word choice.
Page 8: The following sentence is confusing and needs further explanation/clarity: “The water content in the membranes was expressed as a degree of swelling of 473%, which is equivalent to around 80% when expressed as water content on a hydrated basis”
Page 9: The phrase, “Second, the authors used, inexplicably, a wrong equation where A=2.667 [55], while further stating that calculation was done according the international standard . . .” is needlessly condescending with the use of “inexplicably.”
Page 11: providing some explanation on coulometric diffusion would be helpful.
Page 11: The phrase, “Such a large discrepancy can only suggest erroneous results. For chitosan as such, a value of 22 Barrer was measured in the same conditions” is not scientific in that experimental design/apparatus seems more likely to account for the discrepancies than simply, and condescendingly, assuming an error.
Page 13: “ability of glutaraldehyde “to create more empty spaces within the membrane” [76], a rather obscure statement suggesting that the authors may possess an incomplete understanding of the mechanism of crosslinking processes in polymers.” This statement again has an unscientific and needlessly condescending tone. Further, glutaraldehyde is commonly used to form covalent hydrogels across a broad range of biopolymers.
Page 14: “The use of an exaggerated number of units different from the Barrer, sometimes bizarre hybrids of SI, imperial and US units . . .” Inappropriate adjectives.
Reviewer 2 Report
I have reviewed a manuscript entitled “Oxygen Permeability of Silk Fibroin Hydrogels and Their Use as Materials for Contact Lenses: A Purposeful Analysis”. This work aimed to provide a compressive review on the oxygen permeability of BMSF for contact lens application. This work interestingly elaborates the challenges associated with the oxygen permeability of BMSF hydrogels according to the published literature, with a special focus on the contact lens application. I think the layout of the paper is logical and the provided discussion is satisfactory. The novelty of this paper is that the author focused on a certain problem of BMSF with deep insight, that might want others to address this issue before further investigations. In this sense, I think the manuscript worth to be published in the present form that can be of particular interest for the researchers to the field.
Round 2
Reviewer 1 Report
The article revisions adequately address the issues of tone and language raised in the first review. This review will provide a focused and useful assessment of silk fibroin for use in contact lenses.